# Preparation of Boron Nitride Nanoplatelets via Amino Acid Assisted Ball Milling: Towards Thermal Conductivity Application

**DOI:** 10.3390/nano10091652

**Published:** 2020-08-22

**Authors:** Nan Yang, Haifeng Ji, Xiaoxia Jiang, Xiongwei Qu, Xiaojie Zhang, Yue Zhang, Binyuan Liu

**Affiliations:** 1Hebei Key Laboratory of Functional Polymers, Department of Polymer Materials and Engineering, Hebei University of Technology, 8 Guangrong Street, Tianjin 300130, China; 201721504016@stu.hebut.edu.cn (N.Y.); 201911501019@stu.hebut.edu.cn (H.J.); xwqu@hebut.edu.cn (X.Q.); 2School of Chemical Engineering and Technology, Hebei University of Technology, 8 Guangrong Street, Tianjin 300130, China; jiangxx@hebut.edu.cn

**Keywords:** amino acids, hexagonal boron nitride nanoplatelets, mechanical exfoliation, self-healing, thermal conductivity

## Abstract

Hexagonal boron nitride nanoplatelets (BNNPs) have attracted widespread attention due to their unique physical properties and their peeling from the base material. Mechanical exfoliation is a simple, scalable approach to produce single-layer or few-layer BNNPs. In this work, two amino acid grafted boron nitride nanoplatelets, Lys@BNNP and Glu@BNNP, were successfully prepared via ball milling of h-BN with L-Lysine and L-Glutamic acid, respectively. It was found that the dispersion state of Lys@BNNP and Glu@BNNP in water had been effectively stabilized due to the introduction of amino acid moieties which contained a hydrophilic carboxyl group. PVA hydrogel composites with Lys@BNNP and Glu@BNNP as functional fillers were constructed and extensively studied. With 11.3 wt% Lys@BNNP incorporated, the thermal conductivity of Lys@BNNP/PVA hydrogel composite was up to 0.91 W m^−1^K^−1^, increased by 78%, comparing to the neat PVA hydrogel. Meanwhile, the mechanical and self-healing properties of the composites were simultaneously largely enhanced.

## 1. Introduction

Hexagonal boron nitride nanoplatelets (BNNPs), which feature a layered structure analogous to graphene [1,2,3], have gained increasing attention, as they share many of the advantageous properties of graphene, but also show other excellent properties, including highly thermal and chemical stability [4,5,6], excellent mechanical strength [7], and high thermal conductivity [8]. BNNPs consist of sp^2^-bonded atomic layers where B and N atoms are alternately arranged into a hexagonal lattice. The regular atomic arrangement permits high phonon velocity and low phonon scattering, leading to an excellent thermal conductive property (~2000 W m^−1^K^−1^) [9]. Compared to graphene, unique insulating properties make BNNPs especially promising in thermally conductive yet electrically insulating polymer composites [10,11]. However, the preparation of BNNPs still faces a lot of challenges, such as high energy consumption, low yields of production and poor repeatability [12,13]. Recently, some improved methods have been reported to produce BNNPs by exfoliation of hexagonal boron nitride (h-BN) via mechanical cleavage [14]. Up to now, a large number of methods have been explored for the exfoliation process of h-BN. For example, Lin [15] et al. have reported that ball-milled h-BN, where defects had been introduced intentionally, could be further functionalized with a long alkyl chain amine. Chen [16] and his colleagues present a simple and efficient one-step method for the preparation of functionalized few-layer BNNPs by ball milling of commercially available h-BN and urea powder. Among the various mixtures of methods, amino groups play a critical role, since they can produce Lewis acid-base interactions with the electron-deficient boron atoms on the h-BN surface or defects during ball milling [17,18].

Recently, amino acids (AAs) have attracted considerable interest because they have the advantage of origin from biomass sources and are cheap and readily available [19,20]. Owing to their multifunctional groups, AAs have also been widely used in constructing polymer materials with various properties [21,22]. Considering the inherent existence of amino groups, and inspired by the unique structure of AAs, which consist of amino and carboxyl groups and various R-side-chains, we proposed that we could employ the AAs as the ball-milling reagent to exfoliate h-BN. With this approach, h-BN could be exfoliated to BNNPs via Lewis acid-base interactions between the amino groups and boron atoms on the surface of BNNPs, meanwhile carbon-chains, carboxyl and different functional R-side-chains could be grafted onto BNNPs simultaneously. The introduction of amino acids can not only improve the compatibility between the BNNPs and the polymer substrate owing to the carbon-chain moieties, but also endow BNNPs with multi-functions, such as hydrophilicity and hydrogen-bonding interactions, which are attributed to the presence of carboxyl groups and various R-side-chains.

Herein, we made a preliminary attempt to discuss the feasibility for improved mechanical exfoliation of h-BN by using an L-Lysine and L-Glutamic acid-assisted ball-milling process, forming Lys@BNNP and Glu@BNNP. The chemical structures of both Lys@BNNP and Glu@BNNP were proved by specific structure characterization, such as FT-IR and XPS. The results of TEM, SEM and XRD showed that BNNPs with a thinner layer thickness than that of h-BN were obtained, which indicated successful exfoliation. We believe that amino acid-assisted ball-milling exfoliation of boron nitride provides a low-cost, effective and easy-to-operate method for the preparation of multifunctional BNNPs. Furthermore, PVA hydrogel composites with Lys@BNNP and Glu@BNNP as functional fillers have been constructed and studied for highly thermal conductivity and mechanical properties. 

## 2. Materials and Methods 

### 2.1. Materials

L-Lysine, L-Glutamic acid and PVA (1799) were purchased from Aladdin reagent Co., Ltd. (Shanghai, China). Hexagonal boron nitride (h-BN) with lateral particle size in the range of 3–5 μm was purchased from Shandong Qingzhou Matekechuang Materials Co., Ltd. (Shandong, China) Sodium hydroxide, borax and ethanol were purchased from Fuchen Chemical Reagent Co., Ltd. (Tianjin, China). HCl (37%) were provided by Tianjin keraisi Fine Chemical Co., Ltd. (Tianjin, China).

### 2.2. Experiment

#### 2.2.1. Exfoliation of BNNP by H_2_O

A high energy ball mill (JX-5G) was utilized for mechanical exfoliation. At room temperature, 40 g of micron-sized h-BN powder (used as received) and 80 g H_2_O were loaded into a 3 L ball mill tank with 2 kg zirconia ball bead (8 mm:5 mm:3 mm = 1:2:1). The rotational speed of the energy ball mill was set to 350 rpm, and the mixture was milled for 10 h. After cooling back to room temperature, the exfoliation product was collected and rinsed with large amounts of deionized water and ethanol. Followed by filtration and drying in a vacuum oven (60 °C), BNNP was obtained as a white powder with a yield of 92%.

#### 2.2.2. Exfoliation of h-BN by L-Lysine Assisted Ball Milling (Lys@BNNP) 

A high energy ball mill (JX-5G) was utilized for exfoliation. At room temperature, 40 g of micron-sized h-BN powder (used as received), 80 g of L-Lysine and 365 mL of 1.5 M NaOH solution (for the protection of the ball milling equipment) were loaded into a 3 L ball mill tank with 2 kg zirconia ball beads (8 mm:5 mm:3 mm = 1:2:1). The rotational speed of the energy ball mill was set to 350 rpm, and the mixture was milled for 10 h. After cooling back to room temperature, the exfoliation product was collected and rinsed with large amounts of deionized water and ethanol. Followed by filtration and drying in a vacuum oven (60 °C), Lys@BNNP was obtained as a white powder in a yield of 85%.

#### 2.2.3. Exfoliation of h-BN by L-Glutamic Acid Assisted Ball Milling (Glu@BNNP)

The procedure for the synthesis of Lys@BNNP was followed to prepare Glu@BNNP from h-BN and L-Glutamic acid as a white powder with a yield of 83%. 

#### 2.2.4. Preparation of Pure PVA Hydrogel

The route to PVA hydrogel is similar to the previous work [23]. In this experiment, PVA (1799) powder was first dissolved in hot water (98 °C) to form a 12.5 wt% transparent solution. Using borax as a crosslink reagent, PVA/borax solution (40 mL PVA solution and 15 mL 0.06 M borax aqueous solution) was stirred in ultrasonic dispersion for 1 h. Finally, the PVA hydrogel was placed in a cylindrical mold, and a heavy object (~5.0 kg) was placed on the surface of the hydrogel. The purpose of pressing with a heavy object was to remove bubbles and repair cracks in the hydrogel.

#### 2.2.5. Preparation of h-BN/PVA, BNNP/PVA, Lys@BNNP/PVA and Glu@BNNP/PVA Hydrogels

In this experiment, PVA powder was first dissolved in hot water (98 °C) to form a 12.5 wt% transparent solution. Using borax as crosslink reagent, PVA/borax solution (40 mL PVA solution and 15 mL 0.06 M borax aqueous solution) was mixed with a designated amount of h-BN, BNNP/PVA, Lys@BNNP and Glu@BNNP, respectively, under stirring ultrasonic dispersion for 1 h. Finally, the obtained hydrogels were placed in a cylindrical mold and a 5.0 kg pressure was applied for 2 h to remove the interior bubbles.

### 2.3. Characterization

#### 2.3.1. Structural Characterizations

Fourier transform infrared spectroscopy (FT-IR) spectroscopy of h-BN, BNNP, Lys@BNNP and Glu@BNNP were recorded on a Vector-22 (Brucker, Germany) spectrometer. The spectral range was 400 to 4000 cm^−1^ with a resolution of 4 cm^−1^. All spectra were modified with carbon removal and baseline correction. Thermal gravimetric analysis (TGA) was conducted to evaluate the grafting content by using a SDT-TG Q600 thermogravimetric analyzer (TA Instruments, New Castle, DE, USA). The nitrogen flow rate was 100 mL min^−1^, the heating rate was 10 °C min^−1^, and the temperature scanning range was 100–700 °C. X-ray diffraction (XRD) was used to study the crystallinity degrees of the h-BN, BNNP, Lys@BNNP and Glu@BNNP. XRD data was recorded on a D8 Advance (Brook AXS, Germany) X-ray diffractometer. The elemental composition of Lys@BNNP and Glu@BNNP was analyzed by ESCALAB 250Xi (Thermofisher Scientific, Waltham, MA, USA) X-ray photoelectron spectroscopy (XPS). The morphologies of h-BN, BNNP, Lys@BNNP and Glu@BNNP were observed by a Nova Nano SEM 450 (FEI, Hillsboro, OR, USA) scanning electron microscope (SEM). The morphology of the samples was investigated using transmission electron microscopy (TEM) (FEI Talos F200S, Hillsboro, OR, USA). Elemental mapping analysis was conducted by using a FEI Talos F200S microscope. 

#### 2.3.2. Performance Characterizations

The mechanical properties of the nanocomposites were characterized with a dynamic mechanical analyzer (DMA2000B, Tritec Instruments, York, UK) at an oscillation frequency of 1 Hz. The rectangular samples (35 mm × 14 mm × 4 mm) were cooled to −50 °C under liquid nitrogen first, followed by heating up to 70 °C at the rate of 2 K min^−1^.

Thermal conductivity test: The thermal conductivity of all PVA hydrogel samples were measured using TC 3000 Series Thermal Conductivity Apparatus (Xi’an Xiatech Electronic Technology Co., Xi’an, China) by a transient hot-wire technique. In this experiment, a thin metallic wire, as a line heat source and temperature sensor, is suspended between two identical hydrogel samples (6.5 cm × 3.5 cm × 0.3 cm). When the wire temperature is raised by a constant heat flux, it conducts heat to the surrounding medium, i.e., testing the samples. The rate of heat transfer associated with the thermal conductivity of the surrounding materials is reflected by the temperature changes of the wire [24]. Thus, the thermal conductivity *λ* of the tested hydrogels can be calculated by the following formula: $$\lambda =\frac{q}{4\pi \left(d\;\Delta T/d\;\ln t\right)}$$
where, *q* is the heat generation per unit time per unit length of the wire, ∆*T* is the temperature change of the wire, and *t* is the testing time.

Self-Healing Experiments: Hydrogel splines were prepared on a template of 10 cm × 1 cm × 0.4 cm. During the healing process, there was no other stress or outside stimulus applied. After self-healing, the tensile test was carried out again to calculate the healing efficiency [25].

Mechanical Measurements: The mechanical properties of hydrogels were determined by CMT6104 microcomputer controlled electronic universal testing machine. The gravitational sensor of the drawing machine was 100 N. The hydrogel spline was prepared on the template of 10 cm × 1 cm × 0.4 cm. In order to test the rapid self-healing performance of the BN samples, self-healing tests were carried out directly on the testing machine. Firstly, the spline was cut into halves, and then the two separate parts were re-contacted softly without other stress or outside stimulus applied. After waiting for predetermined seconds, the tensile test was carried out immediately. In this case, the healing time was determined from the re-contact to the pressing of the start button. The tensile rate was 100 mm min^−1^, and the initial gauge distance was 60 mm. The data were the average strengths of five samples in the same group [23].

## 3. Results and Discussion

### 3.1. Characterizations of AA@BNNPs

Figure 1 illustrates the overall procedure of the preparation of (amino acid)-*g*-BNNP hybrids (AA@BNNPs) and related AA@BNNP/PVA hydrogels. The Lys@BNNP and Glu@BNNP were prepared via ball milling with L-Lysine and L-Glutamic acid as assisting reagent, respectively, and fully characterized by FT-IR, TGA, XRD, TEM and XPS.

The structural differences of the raw h-BN and the AA@BNNPs were characterized by FT-IR, TGA, XRD and XPS. Figure 2a shows the FT-IR spectra, compared with h-BN, besides B-N bending and stretching vibrations (817 cm^−1^ and 1378 cm^−1^) [26,27], the spectra of BNNP exhibited new absorption bands around 3400~3600 cm^−1^, corresponding to O-H, and the spectra of AA@BNNPs exhibited several new absorption bands around 3400~3600 cm^−1^, 2848 cm^−1^ and 2917 cm^−1^, and 1300~1500 cm^−1^, corresponding to O-H and N-H, C-H, and C=O stretching vibrations, respectively. This result indicated that amino acid moieties had been successfully grafted onto the surface of h-BN, since the unreacted amino acid was removed by repeated washing with DI water and ethanol. TGA measurements provided further evidence of the functionalization of BNNP and AA@BNNPs as shown in Figure 2b. For the Glu@BNNP and Lys@BNNP, a slight mass loss (1.0% and 3.1%) was observed, which was due to the removal of the grafted amino acid moieties. This meant that the grafted quantity of AA@BNNPs was correlating with the number of amino groups. As for BNNP, the mass loss was less (0.4%) due to the removal of hydroxyl groups.

The X-ray diffraction (XRD) of h-BN, BNNP, Lys@BNNP and Glu@BNNP powders is presented in Figure 2c. Two characteristic diffraction peaks were observed at 26.7° and 42.1° arising from (002) and (100) planes of the h-BN, respectively [16]. Apparently, the amino acid does not change the crystal structure of h-BN. A sharp (002) peak with considerably higher intensity at 2θ = 26.7° was presented in the XRD curve for h-BN. Moreover, the (002) peak of Lys@BNNP and Glu@BNNP showed a markedly reduced intensity and dramatically broadened width which indicated thinner BN layers in the hybrids [28]. The chemical bonding state of the AA@BNNPs was examined using X-ray photoelectron spectroscopy (XPS) as shown in Figure 2d,e. Four obvious peaks were observed in the XPS survey scans of Lys@BNNP and Glu@BNNP, corresponding to O 1s, N 1s, C 1s, and B 1s energy, respectively [11,29]. The B 1s spectra of Lys@BNNP and Glu@BNNP can be quantitatively differentiated into two different boron species. The strong peaks at 190.1 and 190.2 eV were coming from the B-N bond, and the weak peaks at 191.0 and 190.8 eV were ascribed to the B-O bond [11,15,29]. The N 1s spectra of Lys@BNNP and Glu@BNNP can be quantitatively differentiated into three different nitrogen species, which are C-N, B-N and B···N [18,19,26,27,30]. Notably, the emergence of new peaks of C 1s and O 1s were also observed with respect to amino acid moieties, in Appendix A. This observation also indicated that the AAs were successfully grafted onto the surface and edge of AA@BNNPs.

It is recognized that the size of the BNNPs has a strong influence on the mechanical and thermal properties of their macroscopic materials. Scanning electron microscopy (SEM) was employed for a detailed morphology study, as shown in Figure 3a–d. The large size of the raw h-BN material was clearly revealed from SEM observation. The raw h-BN sample showed a smooth surface and the layer thickness was around 200–300 nm (Figure 3a). After ball milling, the thickness of BNNP flakes decreased. Moreover, after fully exfoliating with AAs, there was a distinct change in the BN surface, the thickness of AA@BNNPs decreased to around 90 nm, and the surface became rougher. This result was consistent with the XRD observation.

Transmission electron microscopy (TEM) was used to observe the microstructure and crystallinity of BNNP, Lys@BNNP and Glu@BNNP. In Figure 3e,f and Appendix A, TEM images indicated that the basal plane of the AA@BNNPs remained intact, the structure of the few-layer BN remained ordered after ball milling and suggested that the hexagonal lattices of the AA@BNNPs were not damaged during the exfoliation processes. These results were also confirmed through the Raman spectra shown in Appendix A. A typical Raman signature was found for bulk h-BNs and exfoliated AA@BNNPs as a prominent peak in the region of 1364–1366 cm^−1^. It was seen that the peak intensity of BNNPs was notably decreased compared to its raw material h-BN, indicating a reduction of the weaker interaction between layers and layer numbers [9]. The amino acid-assisted exfoliation had not broken up or created any large holes in the nanoplatelets. Besides, compared to BNNP (Appendix A), the TEM results showed that in Glu@BNNP and Lys@BNNP the lamellae were flat and quite thin, which further demonstrated the successful exfoliation of Glu@BNNP and Lys@BNNP from bulk h-BN. It has been noted that the larger lateral size of the as-prepared BNNP was beneficial for effective thermal and mechanical reinforcements [31]. The presence of thick AA@BNNPs flakes can assist thermal transport in the composite because they are less subject to thermal conductivity degradation due to phonon-boundary scattering, and in the composite h-BN nanosheets can form thermal links that interconnect the thick h-BN flakes [32].

Elemental mapping of Lys@BNNP and Glu@BNNP are shown in Figure 3g,h. C, N, O and B elements were observed on the surface and edge of Lys@BNNP and Glu@BNNP samples, which further demonstrated the effective grafting and confirmed the good distribution of amino acid.

To further evaluate the applications of AA@BNNPs, we investigated the stability of the suspensions in water with different concentrations of h-BN, BNNP, Lys@BNNP and Glu@BNNP contents. Portions of 3 mg, 6 mg and 9 mg of each BN samples were mixed with 10 mL water thoroughly in a vortex shaker for 1 min at room temperature. As observed in Figure 4a–d, after sitting for 24 h, the untreated h-BN and BNNP dispersion either floated on the surface or sank to the bottom. The Lys@BNNP and Glu@BNNP dispersions with different concentrations (0.3, 0.6, 0.9 mg mL^−1^) remained stable and uniform. These results demonstrated that, after grafting of amino acid moieties, the hydrophilic carboxyl groups originating from AAs can help to stabilize the AA@BNNPs flakes in water [10,32]. This characteristic should be of benefit for potential applications of AA@BNNPs in constructing well-distributed hydrogels.

### 3.2. Measurements of PVA Hydrogel

PVA hydrogel composites with h-BN, BNNP, Lys@BNNP and Glu@BNNP as functional fillers have been constructed and studied for highly thermal conductivities, mechanical properties and self-healing behaviors. Figure 5a shows that the thermal conductivities of the PVA hydrogel composites gradually increased with the increase of filler content, with respect to the pure PVA hydrogel (0.51 W m^−1^K^−1^). It was found that with 11.3 wt% AA@BNNPs hybrids incorporated, the thermal conductivity of the Lys@BNNP/PVA hydrogel composite was up to 0.91 W m^−1^K^−1^, increased by 78% comparing to the neat PVA hydrogel, as well as Glu@BNNP/PVA with a result of 0.87 W m^−1^K^−1^. Notably, the filling amount of h-BN and BNNP in h-BN/PVA and BNNP/PVA hydrogels cannot reach 11.3 wt% due to its bad dispersion stability in water, as shown in Figure 4a,b. With a similar filling amount (8.3 wt%), Lys@BNNP/PVA (0.83 W m^−1^K^−1^) and Glu@BNNP/PVA (0.82 W m^−1^K^−1^) also exhibited a higher thermal conductivity than that of h-BN/PVA (0.76 W m^−1^K^−1^) and BNNP/PVA (0.77 W m^−1^K^−1^), due to the thick AA@BNNPs flakes which assist thermal transport in the composite because they are less subject to thermal conductivity degradation due to phonon-boundary scattering. In the composite h-BN, nanosheets can form thermal links that interconnect the thick h-BN flakes [9], and the hydrogen-bonding interaction between −OH groups on the PVA chain and −COOH groups origin from amino acid grafts, leading to a better compatibility between the PVA matrix and AA@BNNPs [10,33].

In order to confirm the hydrogen-bonding interaction between −OH groups on PVA chain and −COOH groups, we investigated storage modulus (E) curves and loss factor (tanδ) curves of the PVA composites by dynamic mechanical analysis (DMA) [11]. The mechanical properties of the composites were enhanced by improving the interfacial interaction between the AA@BNNPs and polymer matrix, and by the better dispersion of AA@BNNPs within a polymer matrix. The traces of E and tanδ of PVA composites are shown in Figure 5b,c. As illustrated in Figure 5b, the glass transition temperatures (*T*_g_) acquired from corresponding tanδ peaks of PVA, 8.3 wt% h-BN/PVA, 8.3 wt% BNNP/PVA, 8.3 wt% Glu@BNNP/PVA, 8.3 wt% Lys@BNNP/PVA were 5.9, 6.6, 7.8, 8.3 and 9.1 °C, respectively. Compared with the 8.3 wt% Glu@BNNP/PVA and Lys@BNNP/PVA, the 11.3 wt% Glu@BNNP/PVA and Lys@BNNP/PVA have increased to 8.5 and 9.4 °C. It was concluded that as the Glu@BNNP and Lys@BNNP additional amount increased, the hydrogen-bonding interaction was increased. The *T*_g_ of these PVA composites was increased with different fillers, which suggested that the increase of intermolecular interactions was caused by hydrogen bonds [27]. Moreover, the storage modulus was in the order of the PVA < 8.3 wt% h-BN/PVA < 8.3 wt% BNNP/PVA < 8.3 wt% Glu@BNNP/PVA < 11.3 wt% Glu@BNNP/PVA < 8.3 wt% Lys@BNNP/PVA < 11.3 wt% Lys@BNNP/PVA (see Figure 5c). These dynamic properties were dependent on the incorporated fillers, including the hydrogen-bonding interaction between -OH groups on PVA chain and -COOH groups originating from AA moieties [25,34].

The self-healing experiments route and mechanical measurements [23] of the PVA hydrogel were similar to the previous work. In addition to improving thermal conductivity, the hydrogen-bonding interaction also can enhance the mechanical performance of BNNP/PVA and AA@BNNPs/PVA hydrogel composites, as shown in Figure 5d–f. To further assess their self-healing behaviors [25], the tensile test was carried out to evaluate the healing efficiency. After the hydrogel was cut into two pieces, two parts of the gel were put together and re-contacted for 60 s with no other stress or outside stimulus applied, and then the healed hydrogel was stretched to test the tensile strength, examples as Figure 4e. Lys@BNNP/PVA and Glu@BNNP/PVA hydrogels both retained excellent self-healing properties, the tensile stress to 88% for Lys@BNNP/PVA with filling content of 11.3 wt%, and 74% for Glu@BNNP/PVA with filling content of 8.3 wt%. Compared with Glu@BNNP/PVA, the performance of Lys@BNNP/PVA was much better. The reason was that the grafted amino acid moieties of Lys@BNNP were higher than in Glu@BNNP, and there were more hydrogen bonds between the Lys@BNNP/PVA hydrogels. The detailed experimental data (force-displacement curve and stress-strain curve) and process can be found in Appendix A.

## 4. Conclusions

In summary, we have made a demonstration that economical and biobased AAs can be employed as assisting reagents to prepare well-compatible BNNP derivatives via mechanical exfoliation. With this approach, the dispersion state of AA@BNNPs derivatives in water have been effectively stabilized and transferred into a PVA hydrogel matrix. Based on this, PVA-based hydrogel composites have been prepared with Lys@BNNP and Glu@BNNP as fillers. Significant increases for these composites in thermal conductivity and mechanical properties resulted, relative to those of h-BN/PVA or BNNP/PVA composites and pure PVA hydrogel. This scalable and environmentally friendly approach offers future scope for facile production of various functional BNNPs.

## Figures and Tables

**Figure 1 nanomaterials-10-01652-f001:**
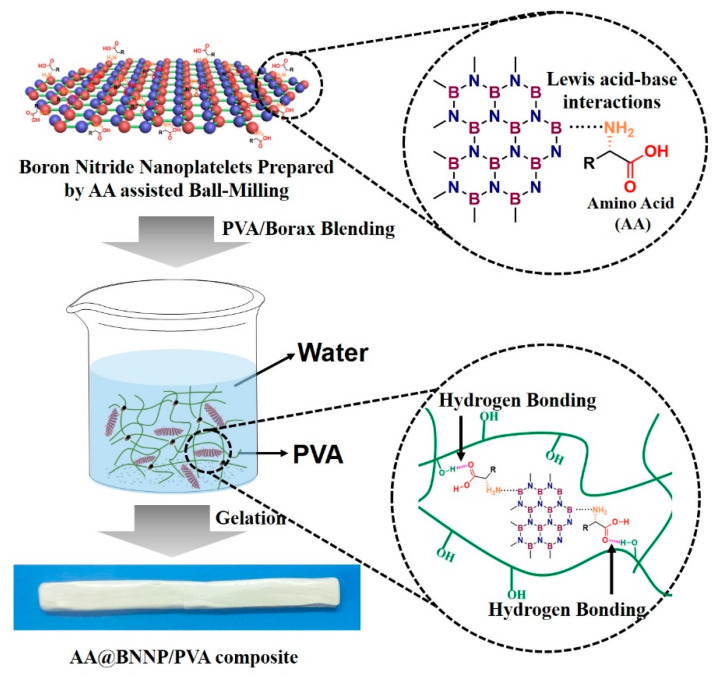
Schematic diagram showing the overall process for preparing the amino acid boron nitride nanoplatelet (AA@BNNP) composites.

**Figure 2 nanomaterials-10-01652-f002:**
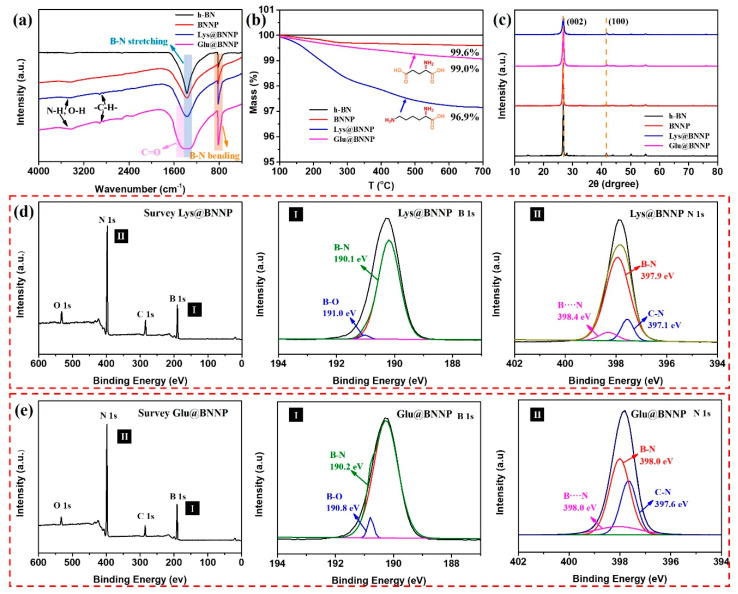
(**a**) FT-IR spectra of h-BN, BNNP, Lys@BNNP and Glu@BNNP; (**b**) TGA curves of h-BN, BNNP, Lys@BNNP and Glu@BNNP under a nitrogen flow; (**c**) XRD patterns of h-BN, BNNP, Lys@BNNP and Glu@BNNP powders; (**d**) high resolution XPS spectra of as-received Lys@BNNP, with (**I**) the B1s XPS spectra of the Lys@BNNP and (**II**) the O1s of the Lys@BNNP; (**e**) high resolution XPS spectra of as-received Glu@BNNP, with (**I**) the B1s XPS spectra of the Glu@BNNP and (**II**) the O1s of the Glu@BNNP.

**Figure 3 nanomaterials-10-01652-f003:**
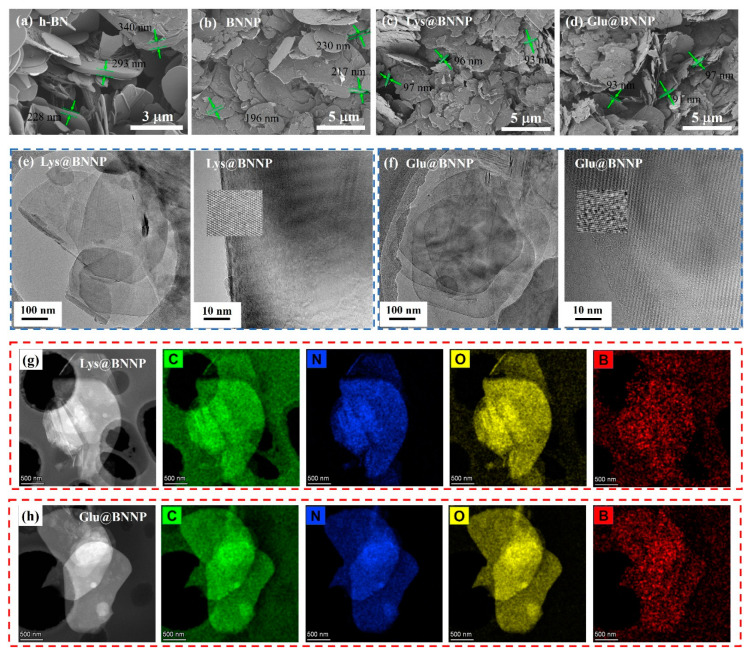
SEM images of (**a**) h-BN; (**b**) BNNP; (**c**) ball milling of Lys@BNNP and (**d**) ball milling of Glu@BNNP; (**e**) low-magnification TEM image of Lys@BNNP and high-resolution TEM images of the Lys@BNNP; (**f**) low-magnification TEM image of Glu@BNNP and high-resolution TEM images of the Glu@BNNP; high-magnification elemental mapping of (**g**) Lys@BNNP and (**h**) Glu@BNNP.

**Figure 4 nanomaterials-10-01652-f004:**
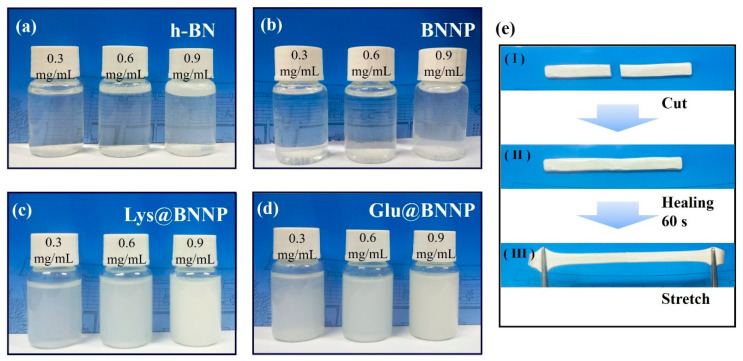
Photographs of as-prepared (**a**) h-BN; (**b**) BNNP; (**c**) Lys@BNNP and (**d**) Glu@BNNP dispersions after 24 h in a concentration of 0.3, 0.6 and 0.9 mg mL^−1^; (**e**) strip-shaped Lys@BNNP/PVA hydrogel with 11.2 wt% of Lys@BNNP: (**I**) the hydrogel is cut into two pieces; (**II**) the hydrogel can heal automatically after re-contacting for 60 s at room temperature; and (**III**) the healed hydrogel is stretched.

**Figure 5 nanomaterials-10-01652-f005:**
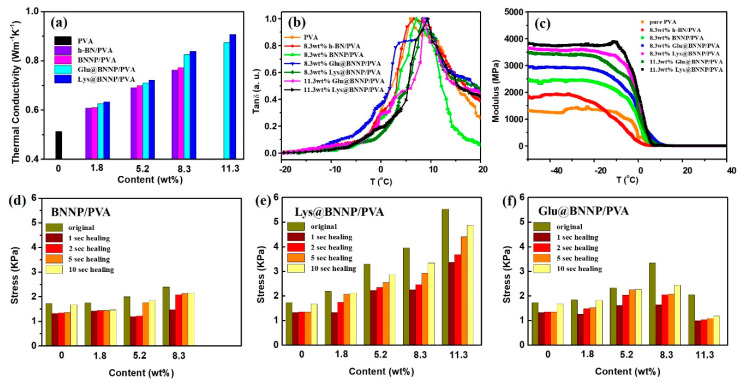
(**a**) Thermal conductivities of pure PVA, h-BN/PVA, BNNP/PVA, Glu@BNNP/PVA and Lys@BNNP/PVA hydrogel; (**b**) dynamic mechanical analysis (DMA) tanδ curves, and (**c**) storage modulus curves of the PVA hydrogels; (**d**–**f**) original stress and healed stress histogram of BNNP/PVA, Lys@BNNP/PVA hydrogel and Glu@BNNP/PVA hydrogel with different contents of BNNP, Lys@BNNP and Glu@BNNP.

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
