# Peer review of "Preparation of Boron Nitride Nanoplatelets via Amino Acid Assisted Ball Milling: Towards Thermal Conductivity Application"

_nanomaterials, 2020, doi:10.3390/nano10091652_

Round 1

Reviewer 1 Report

Your manuscript is well written, the experiments and results are clearly described. I have only a few notes:

  • As you described in experimental part, the milling was done for 10 hours. Please, add the temperature during the milling (or at the end of it).
  • Page 8, line 262: what does it mean 9 in … h-BH flakes,9…
  • Page 8, line 272-273: Tg vales are given with two decimal places, but it does not correspond to obvious error limit of this temperature. Did you determine the reproducibility of this number? Please, give it within the decimal places of error limit and then compare the results.

There are some mistakes as: the units are not according the journal style (no dot should be given); the liter is written as l (not L); the degrees of Celsius are not correct; there should be a space between the number and %; the symbol of thermal conductivity (page 4, line 141) is incorrect as l.

Reviewer 2 Report

This manuscript titled “Preparation of Boron Nitride Nanoplatelets via Amino-acid Assisted Ball-milling: Towards Thermal Conductivity Application” by N. Yang et al reported amino acids-assisted preparation of boron nitride nanoplatelets and their PVA nanocomposite hydrogels. The idea using amino acids is novel. Results and discussion is properly prepared. Therefore, the reviewer recommends the publication in Nanomaterials after following revisions.

  1. Quantitative data; yield of nanoplatelets after a ball milling process, dispersed mass in the aqueous solution (in Fig. 4a-d), are required.
  2. In the part of self-healing experiments; it makes no sense to divide the healing time in seconds. This is because the waiting time for measuring mechanical properties is much longer than 10 seconds. Overall data correction is required.
  3. (L279) Which numerical value is correct? 1.3 or 8.3 ?
  4. (L75) What is the meaning of 1799?
